# A recyclable polyester library from reversible alternating copolymerization of aldehyde and cyclic anhydride

Xun Zhang[1], Wenqi Guo [1], Chengjian Zhang [1] ✉ & Xinghong Zhang [1,2] ✉

Our society is pursuing chemically recyclable polymers to accelerate the green revolution in plastics. Here, we develop a recyclable polyester library from the alternating copolymerization of aldehyde and cyclic anhydride. Although these two monomer sets have little or no thermodynamic driving force for homopolymerization, their copolymerization demonstrates the unexpected alternating characteristics. In addition to readily available monomers, the method is performed under mild conditions, uses common Lewis/Brønsted acids as catalysts, achieves the facile tuning of polyester structure using two distinct monomer sets, and yields 60 polyesters. Interestingly, the copolymerization exhibits the chemical reversibility attributed to its relatively low enthalpy, which makes the resulting polyesters perform closed-loop recycling to monomers at high temperatures. This study provides a modular, efficient, and facile synthesis of recyclable polyesters using sustainable monomers.

Since Hermann Staudinger first proposed the concept of polymerization in 1920[1], polymer science has experienced a hundred years of evolution, promoting polymers to revolutionize our way of life. However, with the vigorous development of the world's modern polymer industry, the contradiction between synthetic polymers and the sustainability of human society is becoming extraordinarily prominent[2–6]. The raw materials for polymer synthesis invariably rely on nonrenewable fossil resources[7–10]. In addition, polyolefins with a carbon-carbon backbone are extremely stable and unexceptionally take hundreds or even thousands of years to degrade into small molecules under natural conditions, leading to grievous environmental concerns[11,12]. In the second century of polymer science, how to strike a balance between polymer materials and social sustainability is one of the top questions.

To address the sustainable and environmental concerns of polymers, recent research has focused on the discovery of renewable or waste carbon resources as raw materials for synthetic polymers that can be biodegraded or recycled at the end of their life[13–20]. Especially chemically recyclable polymers, which can be depolymerized into their starting monomers in a closed loop, not only help address the end-of-life issue of polymers, but also preserve finite natural

resources[3,4,21–37]. Aliphatic polyesters usually possess remarkable biodegradability and biocompatibility and are extensively explored as recyclable plastics[38]. The ring-opening polymerization (ROP) of low-stain lactones is a versatile synthetic method for recyclable polyesters. Besides well-known lactide (LA) and $\varepsilon$-caprolactone ($\varepsilon$-CL), various innovative recyclable lactones have been disclosed recently by Chen[31,32], Hillmyer[39], Lu[26], Tao[24], Hong[23], Zhu[28], Li[40], Xu[33], Tonks[35], Lin[41], and other groups (Fig. 1a). Despite such tremendous progress, available lactones have insignificant diversity and often require multi-step synthesis.

The alternating copolymerization of cyclic anhydride with epoxide is another chain-growth route to prepare polyesters (Fig. 1b)[42–47]. To date, more than 400 polyesters made of more than 20 cyclic anhydrides and 20 epoxides have been reported[48–52]. Cyclic anhydrides are a class of general compounds that can be prepared by heating the corresponding (biorenewable) diacid and losing water in the molecule[48,53,54]. Our group recently reported the alternating copolymerization of cyclic anhydride with cyclic acetal to yield polyesters (Fig. 1b)[55]. Because of the near irreversibility of such copolymerization, the chemical recyclability of the resulting polyesters has not been

[1]National Key Laboratory of Biobased Transportation Fuel Technology, International Research Center for X Polymers, Department of Polymer Science and Engineering, Zhejiang University, Hangzhou 310027, China. [2]Center of Chemistry for Frontier Technologies, Zhejiang University, Hangzhou 310027, China. ✉e-mail: chengjian.zhang@zju.edu.cn; xhzhang@zju.edu.cn

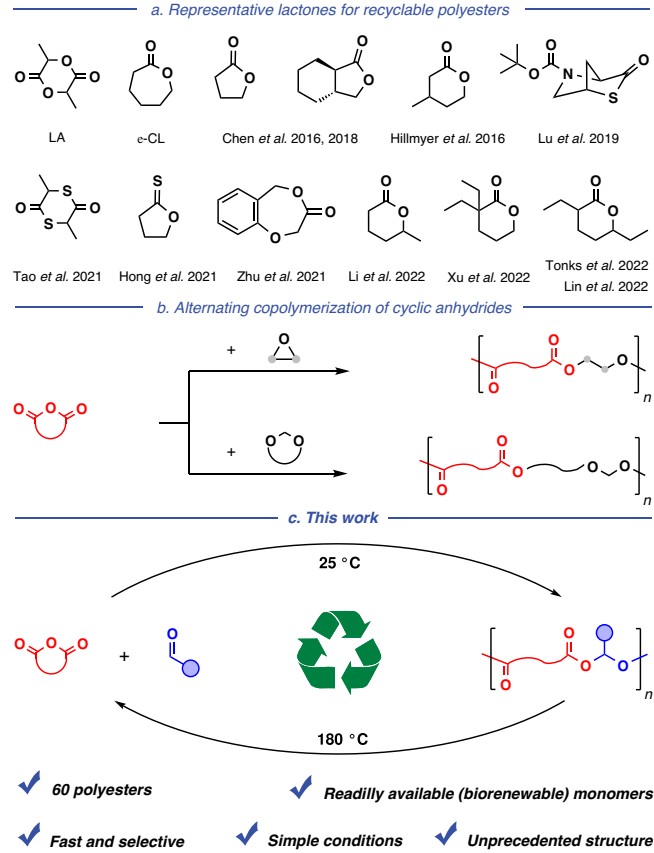

**Fig. 1 | Synthesis of sustainable polyesters. a** Representative lactones for chemically recyclable polyesters. **b** Alternating copolymerization of cyclic anhydride and epoxide/cyclic acetal with near irreversibility. **c** This study: reversible alternating copolymerization of cyclic anhydride and aldehyde to yield polyesters that can be chemically recycled in a closed loop.

demonstrated to our knowledge. It is tremendously challenging to develop chemically recyclable polyesters with easy-to-tune structure and derived from readily available/renewable monomers.

Here, we report the alternating copolymerization of aldehyde and cyclic anhydride with the cationic mechanism (Fig. 1c). Aldehydes are sustainable chemicals with an extensive variety. Plenty of aldehydes can be extracted directly from plants or synthesized by the dehydrogenation of renewable alcohols[56,57]. However, except formaldehyde, other aldehydes are quite difficult to carry out chain (co) polymerization due to their low enthalpy of polymerization according to previous studies[58,59]. In this study, we employed 56 commercially available aldehydes bearing multifarious substituents to deliver a family of polyesters incorporating the aldehyde-anhydride alternating sequence. Of interest, the copolymerization exhibits the desired reversibility owing to the relatively low enthalpy, according to kinetic and thermodynamic studies. Accordingly, these polyesters can be depolymerized to their starting monomers at high temperatures, achieving the closed-loop chemical cycle that is effortless to handle.

## Results

To optimize the copolymerization conditions, we initially used acetaldehyde (**1**) and glutaric anhydride (**A**) as monomers, which can be directly produced from bio-derived ethanol and glutaric acid, respectively. We are encouraged that several commercially available and commonly used Lewis/Brønsted acids in cationic polymerizations[60], including BF$_3$•Et$_2$O, B(C$_6$F$_5$)$_3$, Bu$_2$BOTf, SnCl$_4$, InBr$_3$, NH(OTf)$_2$, and MeOTf are highly effective as catalysts/initiators for **1** and **A** copolymerization (entries 1–7, Supplementary Table 1). At room

temperature and with the feeding ratio [**1**]:[**A**]:[catalyst] = 100:100:1, monomer conversions were in the range of 83 - 90% within 180 s, yielding the copolymer of **P1A** with the number-average molecular weights ($M_n$) of 11.0 - 22.0 kDa and dispersities (Đ) of 1.2 - 1.4, as determined by gel permeation chromatography (GPC). Also, the obtained **P1A** possesses the well-defined aldehyde-anhydride alternating sequence, as determined by NMR spectroscopy (Supplementary Fig. 1).

We then sought to modulate the molecular weight of the copolymer by introducing water as the co-initiator into the BF$_3$-catalyzed **1** and **A** copolymerization. When the water content was improved from 0 to 3, 5, 8, and 10%, the $M_n$ values of the resulting polyesters decreased from 19.7 to 7.1, 5.4, 4.2, and 3.9 kDa, respectively, which are linearly related to [M]$_0$/[H$_2$O]$_0$ (Fig. 2a, b). The addition of 3% water makes the copolymerization rate almost as stable as without adding water. By further increasing the water content to 5, 8 and 10%, the copolymerization rate gradually decreases, reaching a monomer conversion of more than 85% from 180 s without water to about 10 min, 30 min and 2 h, respectively. This phenomenon is because excess water inactivates part of BF$_3$[60]. The obtained polyester of **P1A** possesses high-fidelity carboxyl terminals, as demonstrated by matrix-assisted laser desorption/ionization time-of-flight mass spectroscopy (MALDI-TOF MS). The exclusive series of peaks [HO-(**A**+**1**)$_n$-**A**-H + Na$^+$] belongs to the **P1A** with perfect alternating sequences and double carboxyl terminals (Fig. 2c). We also tested the reactivity of the carboxyl terminals by reacting the **P1A** ($M_n$ = 9.6 kDa) with the diisocyanate at room temperature for 2 h, yielding the polyamide[61] with $M_n$ of 35.1 kDa (Fig. 2d and Supplementary Fig. 2). As a result, the resulting telechelic polyesters have application prospects as reactive macromolecular precursors.

We also investigated the possibility of **1** and **A** copolymerization with the anionic/coordination mechanism. By our experiments, several homogeneous metal and organic catalysts resulted in no products from **1** and **A** copolymerization (entries 8–11, Supplementary Table 1), which have been widely used in epoxide and cyclic anhydride copolymerization[48,62]. This phenomenon is because the active anions are not enough to attack the weaker polar carbon atoms, while the active cations can attack the more polar oxygen atoms to achieve the copolymerization, which is widely recognized in the ROP of heterocyclic compounds[60,63]. Thus, the alternating copolymerization of **1** and **A** was achieved with a cationic mechanism under mild conditions, while neither **1** nor **A** could be homopolymerized under the same conditions (entries 12 and 13, Supplementary Table 1).

Given the spacious diversity of aldehydes, as highlighted in Fig. 3, we then investigated the general applicability of our method to other 55 commercially available aldehydes including aliphatic (**2** to **20**) and aromatic aldehydes (**21** to **56**). Formaldehyde is excluded in this study because it is easy to be homopolymerized via the cationic mechanism. The copolymerization of such aldehydes with **A** was carried out in the presence of 1% BF$_3$•Et$_2$O or InBr$_3$, at 25 or 0 °C, and using CH$_2$Cl$_2$ as the solvent (Supplementary Table 2). All copolymerizations exhibited the superior efficiency, achieving monomer conversions of 61 - 99% within 30 mins. We obtained a series of polyesters of **P2A** to **P56A** with $M_n$ of 2.2 - 16.6 kDa, Đ of 1.2 - 1.6, and the precise alternating sequences determined by NMR analysis (Supplementary Figs. 3–57), suggestive of that our method is resistant of large steric hindrance and common reactive functionalities such as halide, C = C bond, nitro, and cyanogen groups. Also, such reactive groups give the polyesters a lot of room for post-modifications. Prominently, all of these aldehydes can be made from readily available feedstocks, and some of the aldehydes such as **2** to **7**, **10** to **13**, **19** to **21**, and **55** can be extracted directly from plants. The promotion of novel strategies for converting aldehydes into sustainable polymers is eminently luscious but remains challenging[58,64]. In our strategy, the application of such sweeping substrates represents an exceptional polymer synthetic method for assembling aldehydes.

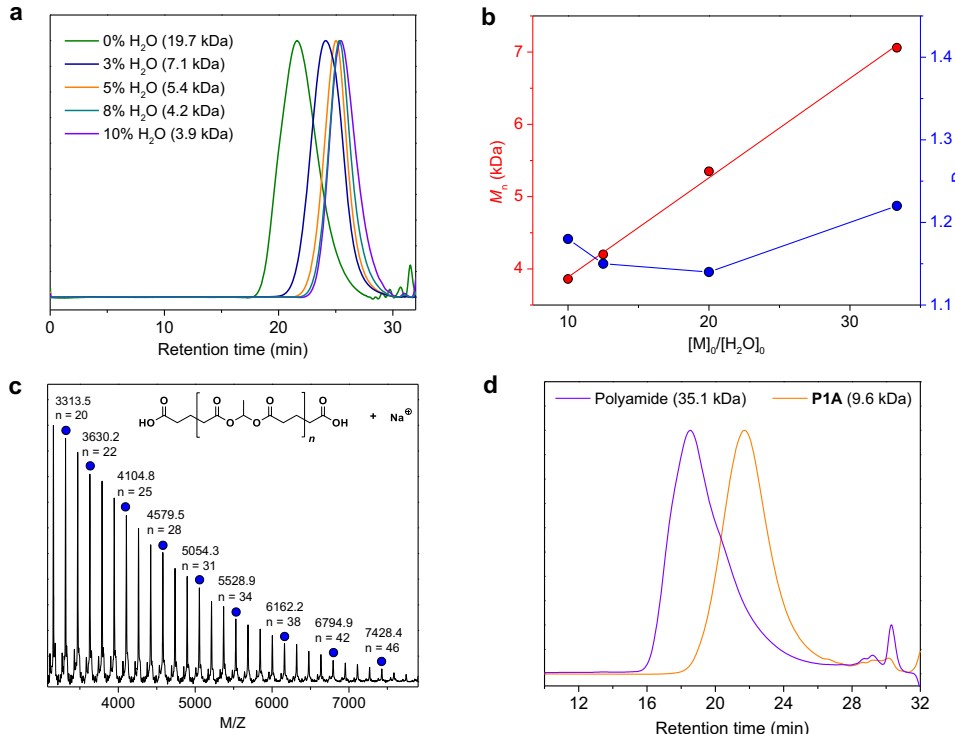

**Fig. 2 | Structure analysis of the resulting** P1A. **a** GPC curves of the obtained **P1A**. **b** Plots of $M_n$ and Đ values versus the ratio of the initial monomer concentration to water concentration ($[M]_0/[H_2O]_0$). **c** MALDI-TOF MS of the **P1A** with low molecular weight. **d** GPC curves of the **P1A** and the formed polyamide.

We also extended our strategy to diverse commercially available six- and seven-membered cyclic anhydrides (**B** to **E**). It's well known that most six-membered cyclic anhydrides are thermodynamically prohibited to homopolymerize by ROP, while the seven-membered cyclic anhydride of **E** can be homopolymerized via the anionic or cationic ROP[65]. In this contribution, under mild conditions (1% BF$_3$·Et$_2$O or InBr$_3$, at 25 °C for 30 mins), the copolymerization of **B** to **E** with **1** manifested the high efficiency and afforded copolymers of **P1B** to **P1E** with $M_n$ of 2.5 ~ 18.3 kDa, Đ of 1.4 ~ 1.5 (Supplementary Table 2), and alternating sequences according to NMR spectra (Supplementary Figs. 58–61). Generally, the synthesis of high-molecular-weight polymers through the cationic polymerization is a major challenge because of the high activity of the cationic propagating species that can cause various side reactions. Interestingly, by our initial explorations, there is no products from the cationic copolymerization of five-membered cyclic anhydrides (maleic anhydride and succinic anhydride) and acetals (**1** and **21**), which could be attributed to thermodynamic prohibitions (Supplementary Table 2). This phenomenon is mainly due to the lower ring tension of the five-membered cyclic anhydride than that of the six-membered cyclic anhydride. By demonstrating the versatility of the huge substrate scope, we produced a polyester library, where two ester bonds are connected face-to-face by a tertiary carbon atom.

We then carried out the kinetic and thermodynamic studies of the aldehyde and cyclic anhydride copolymerization. Interestingly, the copolymerization was determined to be reversible attributed to the relatively low enthalpy, which would be inaccessible in other reported state-of-the-art anhydride-involved copolymerization[48]. We used **1** and **21** as the representative aliphatic and aromatic aldehydes, respectively. For **1** and **A** copolymerization, the equilibrium monomer concentration ($[1]_{eq} = [A]_{eq}$) was measured as a function of temperatures (Supplementary Table 3, Fig. 4a), that are 0.31 M (60 °C), 0.72 M (80 °C), 1.30 M (100 °C), and 2.46 M (120 °C). The Van't Hoff plot of $\ln[A]_{eq}$ versus $1/T$ gave a straight line with a slope of −4.445 and an intercept of 12.203 (Fig. 4b). Accordingly, based on the equation $\ln[A]_{eq} = \Delta H°_{1+A}/RT - \Delta S°_{1+A}/R$, the thermodynamic parameters were calculated to be

$\Delta H°_{1+A} = -36.96$ kJ mol$^{-1}$ and $\Delta S°_{1+A} = -101.46$ J mol$^{-1}$ K$^{-1}$. The ceiling temperature ($T_c$) was calculated to be 141.9 °C at $[1]_0 = [A]_0 = 4.445$ M or $T_c°_{1+A} = 91.1$ °C at $[1]_0 = [A]_0 = 1$ M, based on the equation $T_{c,1+A} = \Delta H°_{1+A}/(\Delta S°_{1+A} + R \ln[A]_0)$, where $R$ is the gas constant. By contrast, by previous studies[66], the thermodynamic parameters of **1** homopolymerization were calculated to be $\Delta H°_1 = -29$ kJ mol$^{-1}$, $\Delta S°_1 = -144$ J mol$^{-1}$ K$^{-1}$, and $T_c°_1 = -71.2$ °C. Therefore, **1** and **A** alternating copolymerization is thermodynamically more favorable than **1** homopolymerization, leading to the alternating feature of the copolymerization.

In addition, by the similar kinetic studies (Supplementary Table 4, Supplementary Fig. 62), **21** and **A** copolymerization demonstrates $\Delta H°_{21+A} = -32.52$ kJ mol$^{-1}$, $\Delta S°_{21+A} = -126.13$ J mol$^{-1}$ K$^{-1}$, and a relatively low $T_c°_{21+A} = -15.3$ °C, while the homopolymerization of **21** or **A** is thermodynamically prohibited. The polymerization heat of **1** and **A** copolymerization is greater than that of **21** and **A** copolymerization. As a result, we conclude that although such aldehydes and cyclic anhydrides used have little or no thermodynamic driving force for homopolymerization, their copolymerization with the alternating characteristics is thermodynamically favorable.

The family of polyesters manifest a broad range of glass transition temperatures ($T_g$) varing from −57 to 73 °C, as determined by differential scanning calorimetric (DSC, Fig. 5a, Supplementary Figs. 63–122). The side chain structure has a great influence on the $T_g$ of such polyesters. Generally, the polyesters bearing aliphatic side chains possess low $T_g$. As examples, the polyesters of **P1A** and **P16A** demonstrate $T_g$ of −15 and −57 °C, respectively. The long carbon side chains enable the polyesters of **P7A** to **P12A** with the semicrystalline characteristics, which exhibit the melting temperatures of −11, 11, 20, 37, 42, and 53 °C, respectively. Additionally, the polyesters of **P18A** to **P20A** bearing alicyclic side chains show higher $T_g$ of 10, 56, and 73 °C, respectively. The aromatic side chains also give the polyesters relatively high $T_g$, such as **P21A** and **P52A** with $T_g$ of 25 and 60 °C, respectively. The resulting polyesters possess $T_d$ (the temperature at polymer decomposition of 5% in mass fraction) values ranging from

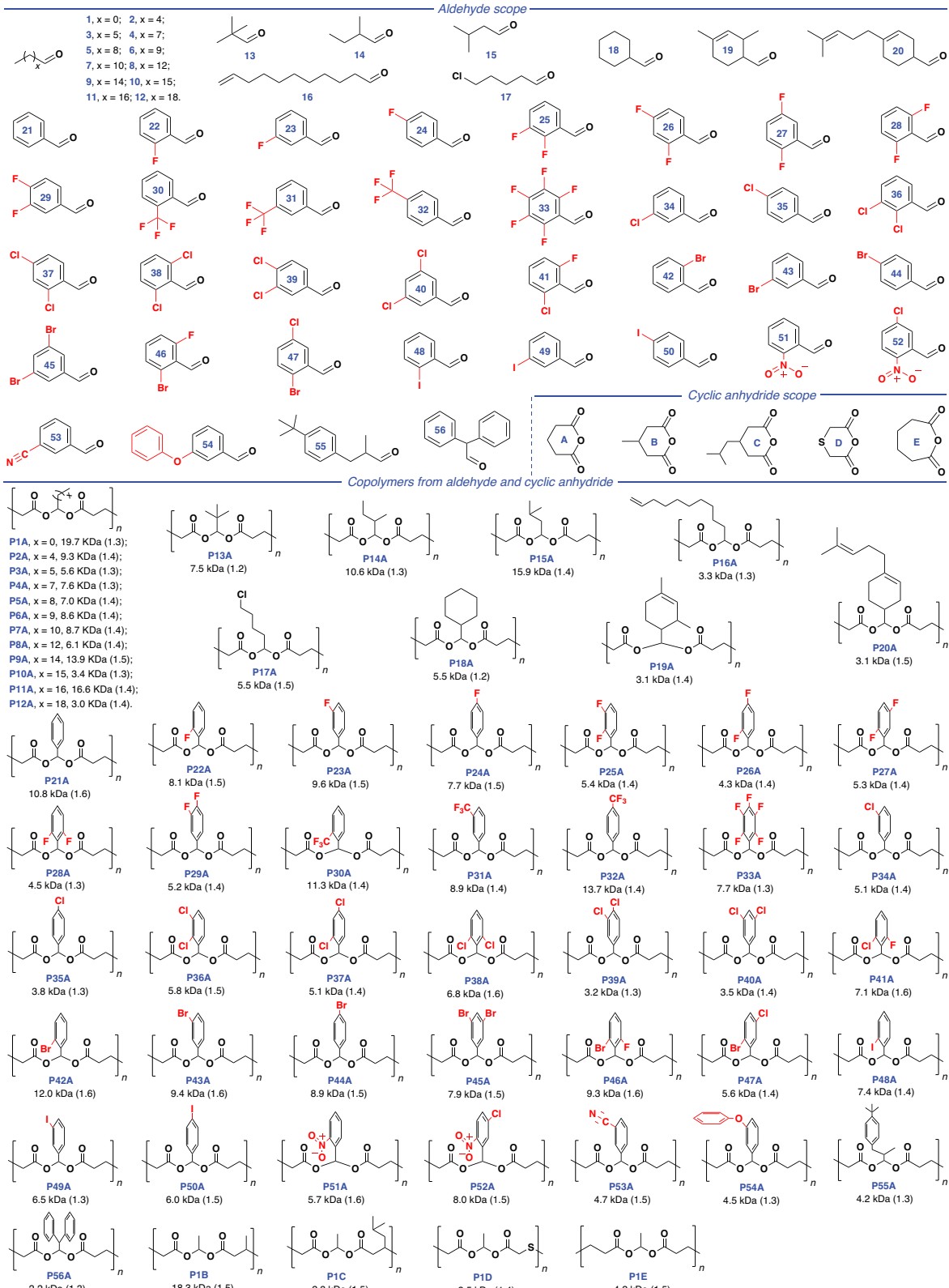

**Fig. 3 | Scope of monomers and obtained polyesters.** $M_n$ and Đ values were determined by GPC in THF, calibrated with polystyrene standards.

170 to 230 °C determined by thermogravimetric (TGA, Supplementary Figs. 63–122). We also tested the mechanical properties of the resulting polyesters taking the high-$T_g$ (50 °C) **P36A** as an example. We successfully synthesized **P36A** on a scale of ten grams by the method (Supplementary Table 5). At −10 °C for 5 h, with the feeding ratio of [**36**]:[**A**]:[InBr₃] = 4000:4000:1, 10.0 g of **36** and 6.5 g of **A** yielded

13.9 g of purified **P36A** with $M_n$ of 18.0 kDa and Đ of 1.5, which is colorless and highly transparent after hot pressing (Fig. 5b). Through the stress-strain experiment at 20 °C, the specimens of **P36A** prepared by the melt processing display the ultimate tensile strength (σ_B) of 3.4 MPa, the elongation at break (ε_B) of 2%, and the calculated Young's modulus of 170 MPa (Fig. 5c). Also, the specimens of **P21A** ($M_n$ of

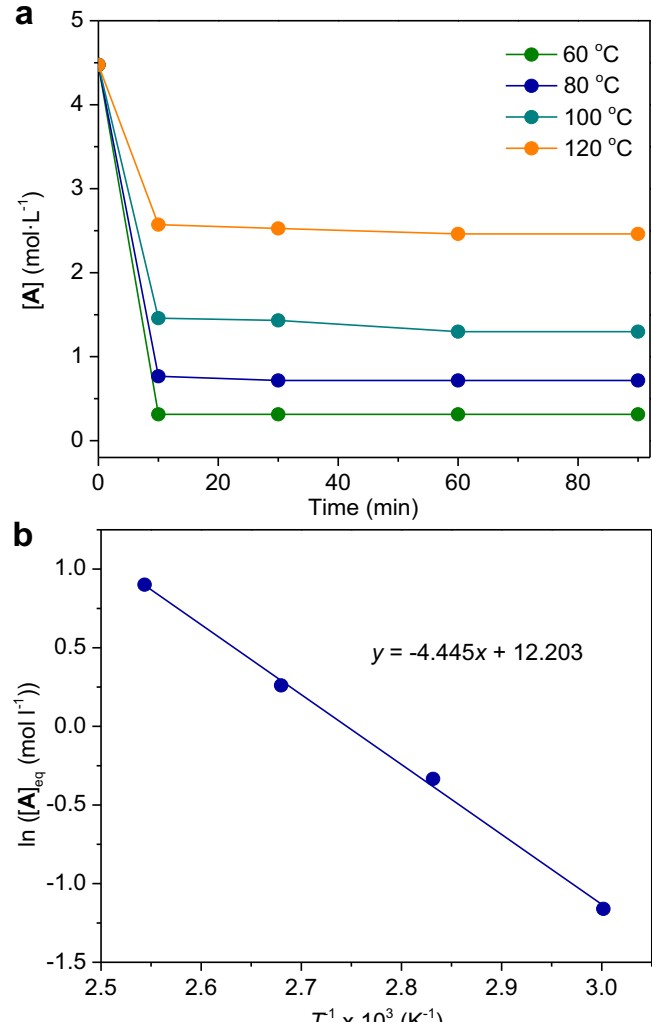

**Fig. 4 | Thermodynamics of 1 and A copolymerization. a** Plot of **A** concentration as a function of time during the copolymerization at different temperatures. **b** Van't Hoff plot of ln[**A**]$_{eq}$ versus the reciprocal of the absolute temperature ($T^{-1}$).

10.8 kDa, Đ of 1.6) exhibit the $\sigma_B$ of 3.1 MPa and the $\varepsilon_B$ of 237% (Fig. 5d). The study of increasing the molecular weight of these polyesters to improve their mechanical properties is necessary in the future.

Based on the reversible feature of the alternating copolymerization, we next sought to test the chemical recyclability of the resulting polyesters with **P36A** as an example. Using a simple commercially purchased sublimation device, at 180 °C for 8 h, under vacuum, in the absence of any solvents and catalysts, 5.0 g of **P36A** was converted into 4.72 g of the mixture of **36** and **A** ([**36**]:[**A**] = 0.94:1, Fig. 5e, f). Without further purification, at −10 °C for 5 h, the subsequent addition of 0.025% InBr$_3$ to the mixture initiated the copolymerization of **36** and **A**, affording 4.15 g of **P36A** with $M_n$ of 17.9 kDa (Đ = 1.5) which is close to that of the original **P36A** ($M_n$ = 18.0 kDa, Đ = 1.5, Supplementary Fig. 123). Consequently, we can realize the closed-loop chemical recycle of the polyester by simple operations.

In conclusion, this study provides a facile and versatile approach to yield recyclable polyesters from abundant renewable feedstocks. We have demonstrated the alternating copolymerization of aldehyde and cyclic anhydride with a cationic mechanism. The method provides a library of chemically recyclable polyesters with tunable structure. The two distinct monomer sets are commercial common chemicals with wide varieties. Besides sustainable monomers, the method is fast, selective, wide

in scope, and carried out under mild conditions. The resulting polyesters possess well-defined alternating sequences, carboxyl terminals, and widely tunable properties. The copolymerization also demonstrates the chemical reversibility as identified by kinetic and thermodynamic studies, which enables the resulting polyesters undergo closed-loop chemical recycling to monomers at high temperatures. Our ongoing efforts are to further clarify the mechanism and develop controllable catalysts for the copolymerization.

## Methods
### Materials
All aldehydes (**1** to **56**) were purchased from Sigma Aldrich Chemical Co. and used as received without further purification. Cyclic anhydrides (**A** to **E**) were purchased from Aladdin Reagent Company (Shanghai) and sublimated twice before use. Catalysts including BF$_3$•Et$_2$O, B(C$_6$F$_5$)$_3$, Bu$_2$BOTf, SnCl$_4$, InBr$_3$, MeOTf, and NH(OTf)$_2$ were purchased from Sigma Aldrich Chemical Co. and used as received.

### Characterization and processing techniques
$^1$H and $^{13}$C NMR spectra were performed on a Bruker Advance DMX 400 MHz. And chemical shift values were referenced to the signal of the solvent (residual proton resonances for $^1$H NMR spectra, carbon resonances for $^{13}$C NMR spectra).

The molar mass and polydispersity of polymers were measured by GPC at 40°C using a Waters 1515 isocratic pump, a model 2414 differential refractometer GPC instrument with tetrahydrofuran as the mobile phase and Waters Styragel HR3, HR4, and HR5 7.8 × 300 mm columns. The flow rate of THF was 1.0 mL/min. Linear polystyrene polymers with narrow molar mass distributions were used as standards to calibrate the apparatus.

MALDI-TOF mass spectrometric measurements were performed on a Bruker Ultraflex MALDI TOF mass spectrometer, equipped with a nitrogen laser delivering 3 ns laser pulses at 337 nm. *Trans*-2-[3-(4-*tert*-butylphenyl)-2-methyl-2-propenylidene] malononitrile was used as the matrix. Sodium trifluoroacetate was added for ion formation.

The decomposition temperature of the polymers were determined by using TA Q50 instrument. The sample was heated from 40 to 500 °C at a rate of 10 °C/min under nitrogen atmosphere. Temperature when the mass loss is five percent was taken as $T_{d,5\%}$.

Differential scanning calorimetry measurements of polymers were carried out on a TA Q200 instrument with a heating/cooling rate of 10 °C/min. Data reported are from second heating cycles.

### Representative procedure for copolymerization
All copolymerizations were carried out in the glovebox under a N$_2$ atmosphere unless otherwise specified. A 10 mL vial with a magnetic stirrer was first dried in an oven at 110 °C overnight, and then immediately placed into the glovebox. The copolymerization of aldehydes and cyclic anhydrides described below is taken from entry 1 in Supplementary Table 1 as an example. CH$_2$Cl$_2$ (0.2 ml), glutaric anhydride (2.3 mmol), acetaldehyde (2.3 mmol), and BF$_3$•Et$_2$O (0.023 mmol) were added into the reactor. Then, the vial was sealed with a Teflon-lined cap and removed from the glovebox. The reaction mixture was stirred at 25 °C for 3 min. After that, an aliquot portion was then taken from the crude product for determining the composition of the crude products by $^1$H NMR spectrum. To obtain purified copolymer, the crude product was dissolved in dichloromethane and then precipitated from an ethanol solution containing sodium phenol (2 M). The resulting polymer was dissolved in dichloromethane and then precipitated from ether for three times. Finally, the obtained polymer was dried in a vacuum.

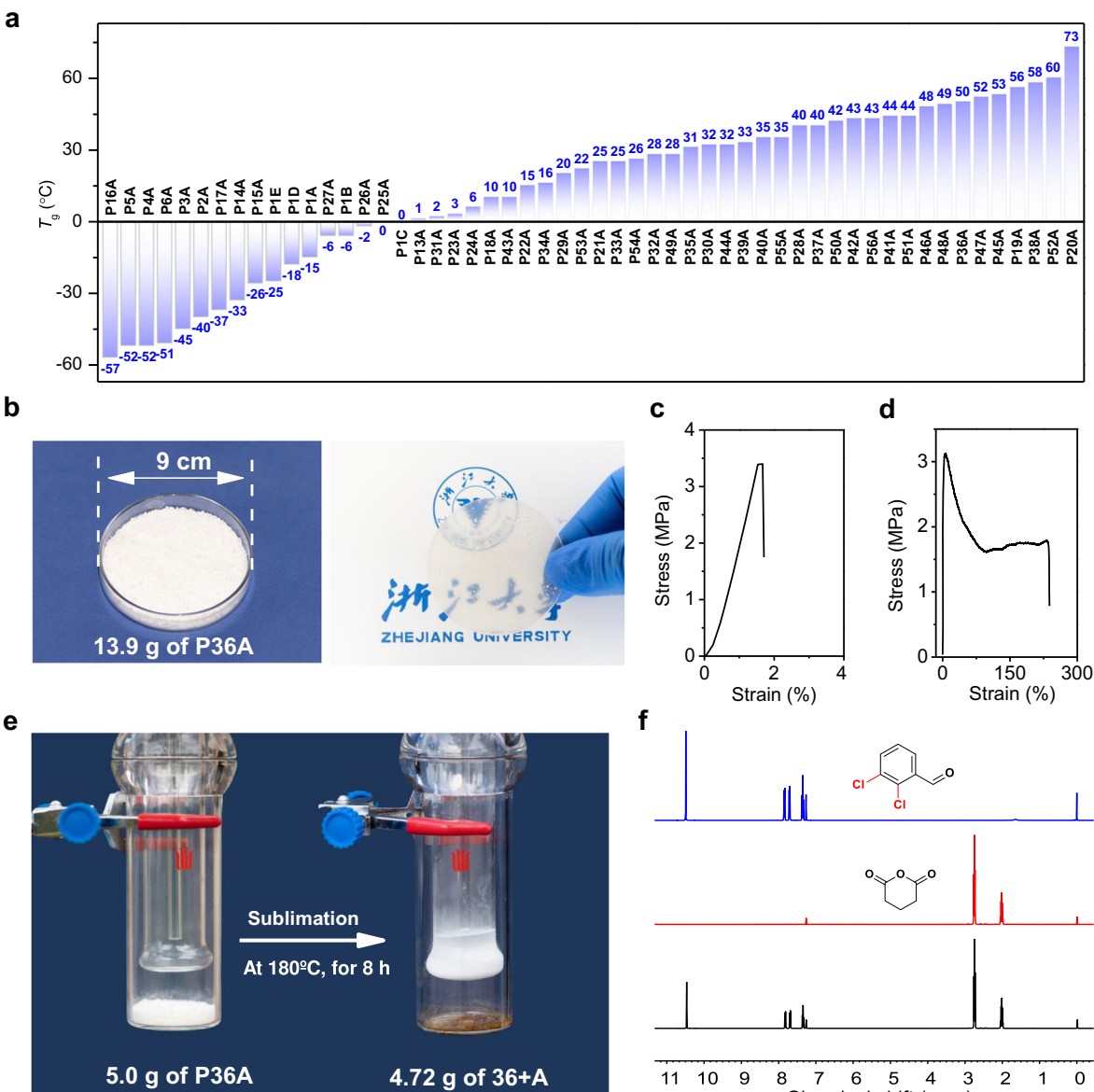

**Fig. 5 | Some properties of the obtained polyesters. a** $T_g$ values of the resulting polyesters. **b** Purified **P36A** and its wafer made from hot press. **c**. Stress-strain curve of **P36A**. **d** Stress-strain curve of **P21A**. **e** Depolymerization of **P36A** to monomers with the sublimation device. **f** $^1$H NMR spectra in CDCl$_3$ of the original **36** (blue line) and **A** (red line) and the regenerated monomers from the depolymerization of **P36A** (black line).

## Data availability

Data supporting the findings of this study are available within the article (and its Supplementary information files). All other data are available from the corresponding author upon request.

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

## Acknowledgements

We acknowledge the financial support of the National Science Foundation of China [no. 52203129 (received by Chengjian Zhang) and 51973190 (received by Xinghong Zhang)] and Zhejiang Provincial Department of Science and Technology (no. 2020R52006, received by Xinghong Zhang).

## Author contributions

X.Z. carried out most of experiments and analysis and wrote the draft. W.G. carried out the analysis of polymer structure. C.Z. conceived, designed, and directed the investigation and revised the manuscript. X.Z. conceived and directed the investigation and revised the manuscript.

## Competing interests

The authors declare no competing interests.
