## [Peer Review File · Nature Communications]

A recyclable polyester library from reversible alternating copolymerization of aldehyde and cyclic anhydrideReviewers' Comments:

Reviewer #1:

Remarks to the Author:

In this manuscript, Zhang and coworkers report the copolymerization of cyclic anhydrides with aldehydes to form linear polymers. They demonstrate the versatility of their polymerization through the polymerization of a very large number of aldehydes and anhydrides. The resulting polymers have glass transitions covering a large range of temperatures. The authors also determine the thermodynamics of two polymerizations. The relatively low heat of polymerization results in a moderate ceiling temperature offering promise for monomer recovery.

Overall, this paper should be published as the discovery of a new polymerization method is truly impactful. The journal should really highlight this work once it is published. The authors did a very diligent job with their experiments, and I have only minor comments.

- In the introduction, the authors talk about polyesters being important and subject to active research. They list a few key recent discoveries but do not mention PLA and PCL, the two large volume polyesters.
 - The water sensitivity is interesting. In absence of water what is initiating the polymerization? Water is a diol, but does it behave like a diol? What is limiting the MW of the polymer?
 - The authors refer to thermodynamic prohibition. I need more details in the manuscript.
 - The manuscript could benefit from some editorial work. The language use is sometimes not the most accurate. While it did not prevent me from understanding the interesting scientific content, I think editorial work will only make the manuscript better.
- Bottomline: Truly impressive and innovative work. Congrats to the authors.

Reviewer #2:

Remarks to the Author:

The paper is comprehensive and contains an impressive amount of work. The concept is quite a simple one (which is good) and the results are important. I think that this could drive future research in the area. The polymers have been characterised by a wide variety of techniques - the Tg plot is figure 5 is very compelling and illustrates a vast array of uses. I very much like the paper and accept.

Reviewer #3:

Remarks to the Author:

This communication describes the use of aldehydes and cyclic anhydrides to produce low ceiling temperature polyesters. The work is innovative from the prior report from the authors published in *Angewandte Chemie* in that at least two of the materials synthesized in this work exhibit observable ceiling temperatures and depolymerize back to monomer and the two-component reaction can be varied to create a library of polyesters with a range of Tg. The impact of the work is likely to spur more investigations and applications of these materials. In the context of the numerous emerging chemically recyclable polymers, however, there is little that makes the work stand out. Thermal properties are modest, decomposition temperatures are low, and mechanical properties are poor (which the authors acknowledge need to be improved). As a reader, it appears the main take-away is that this method is versatile as evident by the large substrate table, but what it teaches the community is unclear beyond adding to the growing list of redesigned polyesters. The work is very well conducted and the conclusions are strongly supported, but I feel is a better fit for another journal.

Some detailed observations are presented for consideration in revisions:

- 1) The scientific work contains a rather large number of subjective terms such as "easy to perform", "unprecedented", "tremendous". I have concerns that sublimation/distillation for a mixture of monomers is "easier" compared to other methods, or that low ceiling temperature polymers could

ever have "tremendous" applications in highly exothermic polyurethane processes. This language is not necessary in my view to convey the science.

2) The selection for prior work to include in Fig 1a is not clear. Lactone systems reported in high impact journals from Prof. Bo Lin and Prof. Ian Tonks are absent in Fig. 1a, although Tonks's work is appropriately referenced, Prof. Lin's is not.

3) Please provide a reference for carboxyl terminated telechelics reacting with isocyanates to produce polyurethanes. While this reviewer is familiar with the formation of polyamides using carboxylates/isocyanates, polyurethanes are typically formed from alcohols/isocyanates, not carboxylic acids. What structural evidence do the authors have for this assignment?

4) Please comment on the observed rate, even if qualitative, upon the addition of water as a co-initiator. I am also not clear what is happening mechanistically in the water initiated system and if the telechelic structure is unique to it or is present in all samples

Reply to Reviewers' Comments

Reviewer 1:

In this manuscript, Zhang and coworkers report the copolymerization of cyclic anhydrides with aldehydes to form linear polymers. They demonstrate the versatility of their polymerization through the polymerization of a very large number of aldehydes and anhydrides. The resulting polymers have glass transitions covering a large range of temperatures. The authors also determine the thermodynamics of two polymerizations. The relatively low heat of polymerization results in a moderate ceiling temperature offering promise for monomer recovery.

Overall, this paper should be published as the discovery of a new polymerization method is truly impactful. The journal should really highlight this work once it is published. The authors did a very diligent job with their experiments, and I have only minor comments.

Reply: Thank you for your encouraging comments.

- In the introduction, the authors talk about polyesters being important and subject to active research. They list a few key recent discoveries but do not mention PLA and PCL, the two large volume polyesters.

Reply: We agree. We have added lactide and ϵ -caprolactone to Figure 1. Also, we have added the description to the revised manuscript: “Besides well-known lactide (LA) and ϵ -caprolactone (ϵ -CL), various innovative recyclable lactones have been disclosed recently by Chen,^{31,32} Hillmyer,³⁹ Lu,²⁶ Tao,²⁴ Hong,²³ Zhu,²⁸ Li,⁴⁰ Xu,³³ Tonks,³³ Lin,⁴¹ and other groups (Fig. 1a).”

- The water sensitivity is interesting. In absence of water what is initiating the polymerization?

Water is a diol, but does it behave like a diol? What is limiting the MW of the polymer?

Reply: Very good questions. Water plays the role of a co-initiator in the polymerization as illustrated in Figure R1. Water first binds to Lewis acid (BF_3) to release protons, and then the protons initiate the copolymerization. As recognized, pure Lewis acid has low initiating activity in cationic polymerizations, and a trace amount of co-initiator is required to ensure the initiation (Kubisa, P.: 4.08 - Cationic ring-opening polymerization of cyclic ethers. In *Polymer Science: A Comprehensive Reference*; Matyjaszewski, K., Möller, M., Eds.; Elsevier: Amsterdam, 2012; pp 141-164). Nevertheless, the mechanism understanding of the uncontrollable cationic polymerization is relatively simple so far.

In our copolymerization system, there is a trace amount of water inevitably, in which cyclic anhydride absorbs water very easily. We believe that the inevitable moisture combined with BF_3 can co-initiate the copolymerization. Compared with diols, water is more acidic. Therefore, a smaller amount of water can ensure high initiator activity, but a large amount of water can cause the Lewis acid inactivated. We propose that in our system, the main factors that limit the molecular weight of the polymer include (1) the presence of trace amounts of water and (2) the high activity of the cationic species leading to the side reaction of transesterification. Our ongoing works will develop controllable catalysts for the synthesis of high-molecular-weight polyesters via our method.

Figure R1. Proposed initiation mechanism for the BF_3 -initiated copolymerization of acetaldehyde and glutaric anhydride.

- The authors refer to thermodynamic prohibition. I need more details in the manuscript.

Reply: We agree. The occurrence of ring-opening (co)polymerization is determined by both kinetics and thermodynamics. For the cationic ring-opening (co)polymerization of heterocyclic compounds, the cationic species is highly active and can easily attack the carbon-hetero bond. Thus, the occurrence of cationic ring-opening (co)polymerization of heterocyclic compounds mainly depends on thermodynamics. In our study, the alternating copolymerization of acetaldehyde and six-membered cyclic anhydrides occurred, while the alternating copolymerization with five-membered cyclic anhydrides did not be triggered. We propose that this phenomenon is mainly due to the lower ring tension of the five-membered cyclic anhydride than that of the six-membered cyclic anhydride. We thus have added the description to the revised manuscript: “Interestingly, by our initial explorations, there is no products from the cationic copolymerization of five-membered cyclic anhydrides (maleic anhydride and succinic anhydride) and acetals (**1** and **21**), which could be attributed to thermodynamic prohibitions

(Table S2). This phenomenon is mainly due to the lower ring tension of the five-membered cyclic anhydride than that of the six-membered cyclic anhydride.”

- The manuscript could benefit from some editorial work. The language use is sometimes not the most accurate. While it did not prevent me from understanding the interesting scientific content, I think editorial work will only make the manuscript better.

Reply: Thank you. We have polished our language in the revised manuscript.

Bottomline: Truly impressive and innovative work. Congrats to the authors.

Reply: Thank you for your encouraging comments.

Reviewer 2:

The paper is comprehensive and contains an impressive amount of work. The concept is quite a simple one (which is good) and the results are important. I think that this could drive future research in the area. The polymers have been characterised by a wide variety of techniques - the T_g plot in figure 5 is very compelling and illustrates a vast array of uses. I very much like the paper and accept.

Reply: Thank you for your encouraging comments.

Reviewer 3:

This communication describes the use of aldehydes and cyclic anhydrides to produce low ceiling temperature polyesters. The work is innovative from the prior report from the authors published in *Angewandte Chemie* in that at least two of the materials synthesized in this work

exhibit observable ceiling temperatures and depolymerize back to monomer and the two-component reaction can be varied to create a library of polyesters with a range of T_g . The impact of the work is likely to spur more investigations and applications of these materials.

Reply: Thank you for your highlight of the innovation of our study.

In the context of the numerous emerging chemically recyclable polymers, however, there is little that makes the work stand out. Thermal properties are modest, decomposition temperatures are low, and mechanical properties are poor (which the authors acknowledge need to be improved). As a reader, it appears the main take-away is that this method is versatile as evident by the large substrate table, but what it teaches the community is unclear beyond adding to the growing list of redesigned polyesters.

Reply: Thank you. Some properties of the resulting copolymer should be improved by our future works. Owing to the versatility of our method and the wide range of monomers, we believe that the structure and properties of the family of polymers can be widely regulated. After this study, our going works will develop more polyesters with desirable performance by design of new aldehydes and cyclic anhydrides as monomers.

The work is very well conducted and the conclusions are strongly supported, but I feel is a better fit for another journal.

Reply: Thank you for your encouraging comments.

Some detailed observations are presented for consideration in revisions:

1) The scientific work contains a rather large number of subjective terms such as “easy to perform”, “unprecedented”, “tremendous”. I have concerns that sublimation/distillation for a

mixture of monomers is “easier” compared to other methods, or that low ceiling temperature polymers could ever have “tremendous” applications in highly exothermic polyurethane processes. This language is not necessary in my view to convey the science.

Reply: We agree and thank you for your kind suggestions. We have deleted such subjective terms and polished our language in the revised manuscript.

2) The selection for prior work to include in Fig 1a is not clear. Lactone systems reported in high impact journals from Prof. Bo Lin and Prof. Ian Tonks are absent in Fig. 1a, although Tonks’ s work is appropriately referenced, Prof. Lin’ s is not.

Reply: We agree. We missed the previous works from Prof. Lin and Prof. Tonks in the previous manuscript. In the revised manuscript, we have added the lactone developed by Prof. Lin and Prof. Tonks in Figure 1. We have also added the following description to the revised manuscript: “Besides well-known lactide (LA) and ϵ -caprolactone (ϵ -CL), various innovative recyclable lactones have been disclosed recently by Chen,^{31,32} Hillmyer,³⁹ Lu,²⁶ Tao,²⁴ Hong,²³ Zhu,²⁸ Li,⁴⁰ Xu,³³ Tonks,³³ Lin,⁴¹ and other groups (Fig. 1a).” Thank you again for your reminding.

3) Please provide a reference for carboxyl terminated telechelics reacting with isocyanates to produce polyurethanes. While this reviewer is familiar with the formation of polyamides using carboxylates/isocyanates, polyurethanes are typically formed from alcohols/isocyanates, not carboxylic acids. What structural evidence do the authors have for this assignment?

Reply: We agree and thank you for pointing out the mistake. The coupling of carboxylates with isocyanates yields the amide group rather than the urethane group as illustrated in Figure R2. The coupling of the carboxylic acid with the isocyanate first produces a thermally unstable dicarboxylic anhydride, and then decomposes to produce amides and carbon dioxide. We also

checked the ^1H NMR spectrum of the resulting polyamide, as shown in Figure R3 (Figure S2 in the revised *Supporting Information*). In the revised manuscript, we have corrected the term of polyurethane with polyamide and also have added a new citation (Sasaki, K.; Crich, D. Facile amide bond formation from carboxylic acids and isocyanates. *Organic Letters* **2011**, *13*, 2256-2259).

Figure R2. The coupling of carboxylates with isocyanates to yield the amide.

Figure R3. ^1H NMR spectrum of the resulting polyamide.

4) Please comment on the observed rate, even if qualitative, upon the addition of water as a co-initiator. I am also not clear what is happening mechanistically in the water initiated system and if the telechelic structure is unique to it or is present in all samples.

Reply: We agree. According to your suggestions, we have added the description to the revised manuscript: “The addition of 3% water makes the copolymerization rate almost as stable as

without adding water. By further increasing the water content to 5, 8 and 10%, the copolymerization rate gradually decreases, reaching a monomer conversion of more than 85% from 180 seconds without water to about 10 minutes, 30 minutes and 2 hours, respectively. This phenomenon is because excess water inactivates part of BF_3 .⁶⁰

Water plays the role of a co-initiator in the polymerization as illustrated in Figure R1. Water first binds to Lewis acid (BF_3) to release protons, and then the protons initiate the copolymerization. As recognized, pure Lewis acid has low initiating activity in cationic polymerizations, and a trace amount of co-initiator is required to ensure the initiation of the copolymerization (Kubisa, P.: 4.08 - Cationic ring-opening polymerization of cyclic ethers. In *Polymer Science: A Comprehensive Reference*; Matyjaszewski, K., Möller, M., Eds.; Elsevier: Amsterdam, 2012; pp 141-164). In our copolymerization system, there is a trace amount of water inevitably, in which cyclic anhydride absorbs water very easily. We believe that the inevitable moisture combined with BF_3 can co-initiate the copolymerization. Therefore, we think that the telechelic structure is present in most of samples.

Figure R1. Proposed initiation mechanism for the BF_3 -initiated copolymerization of acetaldehyde and glutaric anhydride.

Reviewers' Comments:

Reviewer #1:

Remarks to the Author:

I read the response by the authors and approve all the changes made to the manuscript.

Reviewer #3:

Remarks to the Author:

The communication describes a new polymerization method to produce low ceiling temperature polyesters from a diverse subset of monomers. The tunability of thermal properties are demonstrated and the conclusions are well supported. The revised manuscript addresses my concerns and I recommend the work is published after editorial revisions.

Reply to Reviewers' Comments

Reviewer #1 (Remarks to the Author):

I read the response by the authors and approve all the changes made to the manuscript.

Reply: Thank you.

Reviewer #3 (Remarks to the Author):

The communication describes a new polymerization method to produce low ceiling temperature polyesters from a diverse subset of monomers. The tunability of thermal properties are demonstrated and the conclusions are well supported. The revised manuscript addresses my concerns and I recommend the work is published after editorial revisions.

Reply: Thank you for your encouraging comments.